# Enhanced Mouse Susceptibility to Endotoxin Shock after *Plasmodium yoelii* Infection Is Correlated with Increased Serum Levels of Lipopolysaccharide Soluble Receptors

**DOI:** 10.3390/ijms24108851

**Published:** 2023-05-16

**Authors:** Pyone Pyone Soe, Jean-Paul Coutelier

**Affiliations:** 1Unit of Experimental Medicine, de Duve Institute, Université Catholique de Louvain, 1200 Brussels, Belgium; 2Department of Pathology, University of Medicine 1, Yangon 11131, Myanmar

**Keywords:** *Plasmodium yoelii* 265 BY, endotoxic shock, tumor necrosis factor, lipopolysaccharide soluble receptors

## Abstract

Sepsis is a common disease in sub-Saharan Africa and Asia, where malaria is also prevalent. To determine whether *Plasmodium* infection might enhance susceptibility to endotoxin shock, we used a mouse model of lipopolysaccharide (LPS) administration. Our results indicated that *Plasmodium yoelii* infection in mice strongly enhanced the susceptibility of the host to develop endotoxin shock. This increased susceptibility to endotoxin shock was correlated with a synergistic effect of *Plasmodium* and LPS on the secretion of Tumor Necrosis Factor (TNF). TNF contributed mostly to lethality after the dual challenge since neutralization with an anti-TNF antibody provided protection from death. *Plasmodium* infection also induced an enhancement of the serum levels of LPS soluble ligands, sCD14 and Lipopolysaccharide Binding Protein. In this regard, our data confirm that *Plasmodium* infection can profoundly modify responses to secondary bacteria challenges, resulting in dysregulated cytokine expression and pathological effects. If confirmed in humans, LPS soluble receptors might serve as markers of susceptibility to septic shock.

## 1. Introduction

Sepsis is a life-threatening condition that arises when the immune response to a bacterial infection leads to tissue and organ destruction [1]. The World Health Organization has recognized sepsis as a priority for world health [2], responsible for 30 million affected people annually. The disease accounted for 20% of all fatalities worldwide in 2017. These fatalities come primarily from developing countries, with the highest burden in sub-Saharan Africa, particularly its southern and eastern parts, and Asia. More than half of all lethal sepsis cases worldwide are young children, especially neonates [3,4].

The recognition of bacterial components such as lipopolysaccharides (LPS) by the innate immune system is a key factor that triggers the inflammatory host response, leading to the removal of the invasive microorganisms. Signaling occurs after LPS binding to the complex formed by Toll-like receptor 4 (TLR4) and MD-2 on the surface of innate cells such as macrophages and dendritic cells [5]. To be recognized by the TLR4-MD-2 complex, LPS needs to be bound first by Lipopolysaccharide Binding Protein (LBP), a soluble acute-phase protein secreted by hepatocytes [6]. The LPS-LBP complex then binds to CD14, a monocyte differentiation antigen, which allows recognition by the TLR4-MD-2 complex [7]. Interestingly, both a membrane and a soluble form of CD14 exist. During sepsis, an imbalance in cytokine production results in an excessive production of proinflammatory cytokines (Th1 type), with production of proteases and complement activation and impairment of the anti-inflammatory response [8,9]. Dysfunction of the normal balance between cytokines has the potential to amplify a systemic response to infection, including enhanced phagocytic activity, vascular endothelial injury with capillary leak and activation of the coagulation system, leading to hypotension and shock [10,11].

Concomitant bacterial, viral or parasitic infections may potentiate this inflammatory reaction as a result of proinflammatory priming, enhancing the susceptibility of the host to septic shock with multi-organ damage, and increase mortality [10]. Such an exacerbating effect of viral infection on the host’s susceptibility to concomitant endotoxin shock was reported in mice after infection with lactate dehydrogenase-elevating virus (LDV) and *Plasmodium chabaudi* [12,13]. *Plasmodium* infection causes proinflammatory priming and hyper-responsiveness to microbial products in humans [14,15]. In this study, we investigated in a mouse model whether infection with *Plasmodium yoelii* can exacerbate endotoxin shock and whether serum markers for this enhanced susceptibility could be identified.

## 2. Results

### 2.1. Infection and Survival

129/Sv mice were challenged with *Plasmodium yoelii* 265 BY (10^6^ iRBCs) and followed every 3–4 days for parasitemia. Higher parasitemia and anemia were observed between day 12 and day 16 post infection, and mice recovered at day 20 post infection (Figure 1). Although animals showed signs of disease, lethal infection was uncommon. Sensitivity to endotoxin shock was measured at day 10 post infection, at the beginning of the severe phase.

### 2.2. Enhanced Sensitivity to Endotoxin after Plasmodium Infection

129/Sv mice were infected with *Plasmodium yoelii* 265 BY and followed every three days for parasitemia. Although animals showed signs of infection, lethal infection was uncommon. Either noninfected or *Plasmodium*-infected animals were then challenged with LPS 10 days after *Plasmodium* infection and monitored for survival. A high dose of LPS that was found previously to be nonlethal for uninfected 129/Sv mice [12] was used. A strong enhancement of susceptibility to endotoxin shock was observed in *Plasmodium*-infected mice with lethality occurring the first day after LPS administration (Figure 2, *p* = 0.0143, representative of two independent experiments).

### 2.3. Effect of Plasmodium Infection on LPS-Induced Tumor Necrosis Factor (TNF) Release

Since TNF-α plays a major role in the induction of endotoxin shock [16], the consequences of *Plasmodium* infection on its production in response to LPS were analyzed. TNF-α levels were moderately increased in noninfected mice at 2 h after LPS administration (Figure 3). *Plasmodium* infection alone did not induce significant TNF production; however, LPS administration resulted in a strong enhancement of TNF production at two hours after LPS challenge in mice infected for 10 days with *Plasmodium* (Figure 3, *p* < 0.001). This TNF production then decreased rapidly. 

### 2.4. TNFα Involvement in the Exacerbation of Susceptibility to Endotoxin Shock by Plasmodium Infection

To determine the role of TNF in the enhancement of susceptibility to endotoxin shock after *Plasmodium* infection, mice were treated with an anti-TNF polyclonal antibody that was previously shown to neutralize TNF and to inhibit LDV-enhanced endotoxin shock [12]. This treatment resulted in the survival of most *Plasmodium*-infected mice, even after administration of a high dose of LPS (Figure 4, *p* < 0.0001, shown for two combined independent experiments).

### 2.5. Effect of Plasmodium yoelii Infection on Soluble LPS Receptors Production

After exposure to LPS, soluble LPS receptors such as sCD14 and LBP can modulate LPS binding to the cell membrane [17]. Therefore, sCD14 and LBP production were measured in the serum of control and *Plasmodium*-infected 129/Sv mice, 10 days after infection. *Plasmodium yoelii* 265 BY induced an enhancement of both sCD14 (*p* = 0.0274) and LBP (*p* = 0.0002) production (Figure 5, representative of two independent experiments).

## 3. Discussion

Susceptibility to septic shock has previously been reported both in patients and in animal models to be enhanced by various infections, including with influenza virus, respiratory syncitial virus, LDV, lymphocytic choriomeningitis virus (LCMV), adenovirus, Theiler’s virus, vesicular stomatitis virus and *Plasmodium chabaudi* [12,13,18,19,20,21,22]. We confirmed here after *Plasmodium yoelii* infection that this potentiation of the LPS pathogenic effect by infectious agents was unrelated to bacteria. Since experimental *Plasmodium* strains have been shown to be contaminated with LDV [23], which has been shown to increase susceptibility to endotoxin [12], it was important to check for the absence of this virus in our parasite stock to assess whether this exacerbating effect was related to the parasite itself. Indeed, a similar exacerbation of the susceptibility to LPS after infection with a *Plasmodium berghei* strain that was contaminated with LDV did not allow us to determine whether this effect was due to the virus or to the parasite (unpublished data). However, contamination of our *Plasmodium yoelii* strain with another undetected virus cannot be excluded. Severe malaria triggers intestinal injury in children [24], which might lead to bacterial invasion and septic shock. However, our results show an enhanced sensitivity to endotoxin administered independently from the parasite rather than an increased endotoxin load.

TNF production was synergistically enhanced after LPS exposure in mice infected with *Plasmodium yoelii*, similarly to what was previously reported after infection with LDV, adenovirus, LCMV and *Plasmodium chabaudi* [12,13,14,18,19,22]. However, the synergistic enhancement of TNF production after concomitant exposure to LPS and infection with LDV was much more important in vivo than ex vivo, using macrophages derived from LDV-infected mice [12]. This suggested that other cells than macrophages were responsible for a large part of TNF production, and/or that additional in vivo mechanisms enhanced the capacity of macrophages to produce TNF. Whatever the mechanisms leading to this enhanced TNF production, it was responsible for the endotoxic shock, since neutralization of this cytokine prevented mouse lethality. Other cytokines have been reported to be also either increased and/or involved in shock development. After LDV infection, IFN-g has a pathogenic effect, while type I IFN protects against shock [12]. A similar pathogenic effect of IFN-g was reported after LCMV infection [18,19] and was probably related to its ability to enhance TNF production by macrophages. IL-12, which stimulates IFN-g production, notably by NK cells, therefore also plays a central role in the virally enhanced susceptibility to LPS [19,25]. Both IFN-g and TNF are involved in the overproduction of IL-1b that is required for shock induction [13]. In contrast to what is observed after LDV infection, LCMV-induced type I IFN also exacerbates sensitivity to shock [20,21]. This may be explained by opposite effects of type I IFNs on the production of IFN-g. Finally, suppression of or delay in the production of regulatory cytokines such as IL-10 may also be involved in the enhancement of endotoxin susceptibility by concomitant nonbacterial infections [12].

Enhanced TNF production in response to LPS after infection with an infectious agent unrelated to bacteria might result from enhanced TLR4 expression, although such an increased TLR4 expression was at best modest after LDV infection [26]. Although enhanced TLR4 expression was reported on monocytes of malaria patients, in a mouse model TLR4 expression was much less enhanced than that of other TLRs after infection with *P. chabaudi* [14]. In this model, TLR9, whose expression was strongly enhanced after *Plasmodium* infection, was required for exacerbated response to LPS, probably through the production of pro-inflammatory cytokines such as IL-12 and IFN-γ [14].

Soluble LPS receptors, and especially LBP levels, are enhanced after bacterial infections. After infections with HIV and dengue virus, these soluble LPS receptors may be produced as a result of microbial translocation [27,28]. Here, we found an enhancement of both LBP and sCD14 after mouse infection with *Plasmodium yoelii*, at the early phase of increased parasitemia and anemia, although this parasite is not supposed to alter the gut barrier. This enhancement was mostly marked for LBP, an acute-phase protein. LBP levels are also increased in patients with malaria [29]. Interestingly, a similar enhancement of sCD14 and LBP production was found after mouse infection with LDV, a virus that does not affect the gut but that also enhances the susceptibility of its host to endotoxin shock [26]. Although LBP-independent mechanisms may also mediate in vivo cellular responses to LPS [30], LBP usually plays a major role in the transfer of LPS to CD14 and in the resulting cellular response, cytokine production and lethality in mice challenged with endotoxin [8,31]. Depending on its serum concentration, LBP might either enhance LPS pathogenicity or protect against it [32]. On the other hand, sCD14 is a marker of monocyte activation whose production is triggered by proinflammatory cytokines [33]. sCD14 may induce response to LPS, especially when it is associated with LBP, in cells devoid of membrane CD14 [34]. Increased sCD14 levels have been associated with high lethality of septic shock [35]. Since these soluble receptors may, in some circumstances, enhance response to LPS, their role in the increased susceptibility to endotoxin after *Plasmodium* or viral infection may be hypothesized. Further kinetic experiments after various experimental infections measuring both soluble LPS receptor levels and the increased susceptibility of animals to septic shock, as well as clinical studies evaluating the levels of these receptors in patients infected with various pathogens, including *Plasmodium*, that render them more susceptible to septic shock, should provide valuable information on the correlation between shock susceptibility and serum LPS receptor levels. Regardless of a possible causative role in the enhanced susceptibility to septic shock, it might then be possible to determine whether LBP and sCD14 serum levels might be good prognosis markers of such an enhanced susceptibility.

## 4. Materials and Methods

### 4.1. Mice

129/Sv female mice were bred at the Ludwig Institute for Cancer Research by Dr. Pedro Gomez Pinilla and used at the age of 7–9 weeks. This mouse strain was used since it was found to be very sensitive to enhancement of endotoxin shock by infection [12]. The study was performed with the approval of the «Comité d’Ethique facultaire pour l’Expérimentation Animale—Secteur des Sciences de la Santé—Université catholique de Louvain» (ref. 2014/UCL/MD/008 and 2018/UCL/MD/007).

### 4.2. LPS

LPS from Escherichia coli (0111: B4) (Sigma-Aldrich, St. Louis, MO, USA; ref L2630) was administered in sterile PBS by intraperitoneal injection.

### 4.3. Plasmodium Infection

Blood-stage samples of the rodent parasite *Plasmodium yoelii* 265 BY were kept at −80 °C until use. 129/Sv mice were infected by i.p. injection of 1–2 × 10^6^ infected erythrocytes (iRBCs). This *Plasmodium* strain was found to be free of LDV contamination by PCR analysis performed by Charles River Research Animal Diagnostic Services (Wilmington, MA, USA).

Laboratory values that reflect *Plasmodium* infection were obtained by microscopy on thin blood films stained with Giemsa every 3–4 days throughout the infection. 

### 4.4. Cytokine and LPS Soluble Receptor Assays

For TNF, Maxisorb ELISA plates (Nunc, UK) were coated with 4 µg/mL of anti- TNF-α (ref. 14-7325-85). After blocking in PBS with 10% FCS, samples were incubated for 2 h at 37 °C, followed by detection antibodies (4 µg/mL of biotinylated TNF-α Antibody Cocktail (ref. 13-7326-85)), by avidin-HRP (1:2000 dilution; ref. 405103, Biolegend, San Diego, CA, USA), by 1-Step™ Ultra TMB-ELISA (ref. 34028, Thermo Fisher Scientific, Waltham, MA, USA) and by 20 µL of stop solution (2M H_2_SO_4_). Serum soluble CD14 and LBP concentrations were measured by using quantification ELISA kits (Cat No:042 for sCD14 and Cat No:043 for LBP, Biometec GmbH, Greifswald, Germany). All absorbance reads were performed at 450 nm, using a 96-well plate spectrophotometer (VersaMax, Molecular Devices, San Jose, CA, USA).

### 4.5. Antibodies

Goat anti-TNF antiserum was prepared in the Hormonology Laboratory of the Centre d’Economie Rurale (Marloie, Belgium) and polyclonal antibody was precipitated with ammonium sulfate [12]. An amount of 5 mg of polyclonal anti-TNF antibody and the same dose of IgG goat antibody (HIL9K6, obtained from non-immunized goat) as control were given i.p. in 1 mL PBS 1 h before LPS.

### 4.6. Statistical Analysis

Statistical analysis was performed with Prism 6 (GraphPad Software, La Jolla, CA, USA) using non-parametric test (Mann–Whitney), 2-way ANOVA (multiple comparisons) and log-rank test (survival curve). *p* values < 0.05 were considered to be statistically significant.

## Figures and Tables

**Figure 1 ijms-24-08851-f001:**
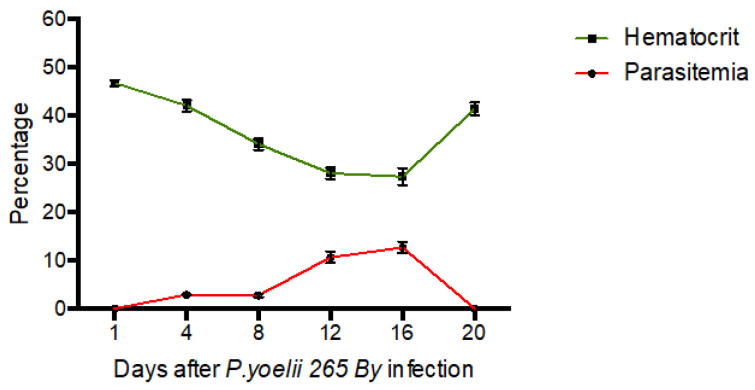
Kinetics of parasitemia and hematocrit in 129/Sv mice after *Plasmodium yoelii* administration. Hematocrit (%, green) and parasitemia (% infected red blood cells, red). Results for 3 mice, as means ± SEM.

**Figure 2 ijms-24-08851-f002:**
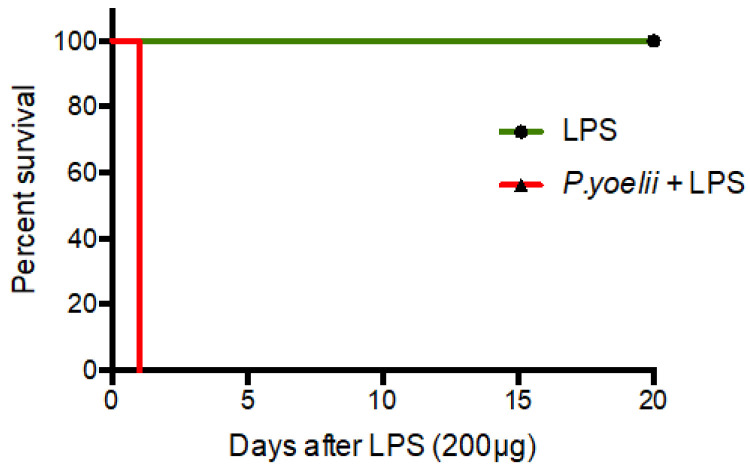
Effect of *Plasmodium yoelii* infection on susceptibility to LPS shock; 129/Sv mice (4 per group), either non-infected or infected with *Plasmodium yoelii* 265 BY for 10 days, were challenged with LPS (200 µg) and monitored for survival.

**Figure 3 ijms-24-08851-f003:**
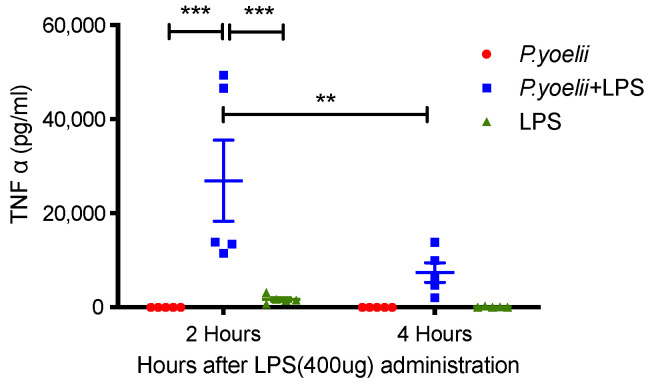
Effect of *Plasmodium* infection on LPS-induced TNF-α release. TNF-α levels were quantitated by ELISA in sera of uninfected or *Plasmodium*-infected 129/Sv mice at 2 and 4 h after LPS administration (400 µg). Results representative of two independent experiments with 4–5 mice per group are expressed as means ± SEM. Statistically significant differences between results comparing uninfected or *Plasmodium*-infected mice challenged with LPS were indicated as *p* < 0.01 (**) and *p* < 0.001 (***).

**Figure 4 ijms-24-08851-f004:**
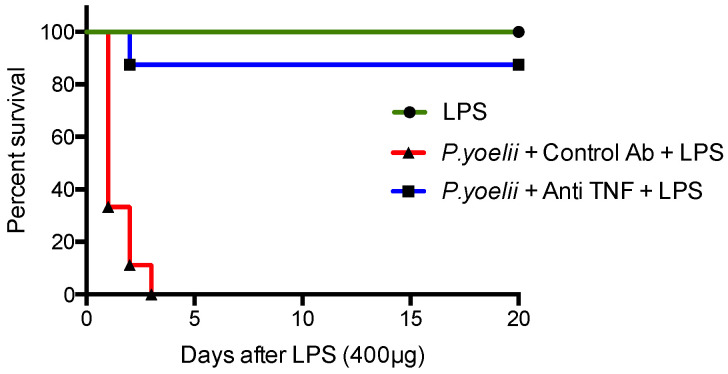
Effect of TNF α neutralization on *Plasmodium*-enhanced susceptibility to LPS. An amount of 5 mg of polyclonal anti-TNF antibody and of control IgG goat antibody were given 1 h before administration of 400 µg of LPS to 129/Sv mice, 10 days after *Plasmodium* infection or in uninfected animals. *p* < 0.0001 when comparing infected mice receiving LPS and treated with anti-TNF versus control Ab. Results are shown for two combined independent experiments with 4–5 mice per group in each experiment.

**Figure 5 ijms-24-08851-f005:**
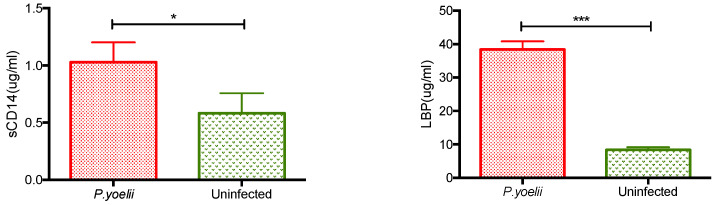
Production of LPS soluble ligands after *Plasmodium* infection. sCD14 and LBP were measured by ELISA in the serum of 129/Sv mice 10 days after injection of either saline (uninfected) or *Plasmodium*. Results for 6–8 mice per group are shown as mean ± SEM. Statistically significant differences between results comparing uninfected or *Plasmodium*-infected mice were indicated as *p* < 0.05 (*) and *p* < 0.001 (***).

## Data Availability

Data are available upon reasonable request.

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
