# Peer review of "Enhanced Mouse Susceptibility to Endotoxin Shock after Plasmodium yoelii Infection Is Correlated with Increased Serum Levels of Lipopolysaccharide Soluble Receptors"

_ijms, 2023, doi:10.3390/ijms24108851_

Round 1

Reviewer 1 Report

The manuscript “enhanced mouse susceptibility to endotoxin shock after Plasmodium yoeli infection is correlated with increased serum levels of lipopolysaccharide soluble receptors” by Pyone Pyone Soe and Jean-Paul Coutelier describes that mice infected with rodent malaria parasite increased susceptibility to endotoxin shock.

This study would have a certain importance, but too preliminary to be published in IJMS.

1, Authors describe malaria parasite used in this study was free of LDV contamination (page5 line169). Show the data about it. Authors cite ref20 for LDV contamination in experimental malaria parasite (page4 line116). However, ref20 did not contain such data.

2, 129/Sv strain is seldom used for rodent malaria infection. Authors should explain why 129/Sv, not BALB/c or C57BL6, more frequently used for rodent malaria infection, was used in this study.

3, In Figure4, “uninfected” should be used instead of “control” in the graph.

4, The dose of LPS varies among the experiments (200mg or 400mg). Authors should explain how the dose of LPS was determined in each experiment. Moreover, it is common to describe the dose of LPS as mg/kg.

5, In this study, administration of 400mg LPS did not induce endotoxin shock in mice. The positive control; administration of enough amount of LPS induces endotoxin shock and death of mice should be included in the data. Based on this, sub-lethal or non-lethal dose of LPS should be determined and shown in this study, which could be examined whether susceptibility to endotoxin shock may be increased or not combined with Py infection.

6, Plasmodium “yoelii”, not “yoeli”.

7, The parasitemia data is completely lacked in this study. The parasitemia growth should be demonstrated. The reader could not understand the mice condition day 10 after Py infection. Authors should explain why day 10 after Py infection was chosen as timing of administration of LPS.

8, Page3 line76-80 should be placed in Figure 2 legend.

9, Authors should explain how to generate goat anti-TNF serum. Authors cite ref9, but it is not adequate. If this antiserum was used in previously published study, cite that. If not, demonstrate that this antiserum is confirmed to neutralize TNF bioactivity in vivo or in vitro. Is 5mg of anti-TNF serum enough to neutralize TNF activity in mice received LPS? What is control antibody (HIL9K6)? prepared from not immunized goat? Which line indicates mice received control antibody in Figure3?

10, Figure4 shows the expression of sCD14 and LBP was increased day10 after Py infection. Why were the expression of these proteins increased? What kind of cells express these proteins? Did authors examine the expression profile (mRNA, protein level) in cells usually expressing these proteins?

11, The increased level of sCD14 and LBP may be linked to increase susceptibility to endotoxin shock, but it has not been experimentally proven in this study. Is the increase level of sCD14 and LBP enough to increase reactivity of LPS to TLR4? How about the expression level of TLR4 in Py infected mice?

Author Response

We thank the reviewer for his.her useful comments. Changes and answers are as follows:

1, Authors describe malaria parasite used in this study was free of LDV contamination (page5 line169). Show the data about it. Authors cite ref20 for LDV contamination in experimental malaria parasite (page4 line116). However, ref20 did not contain such data.

Analysis of LDV contamination was performed commercially by Charles River Research Animals Diagnostic Services. This was added in the MM section. Reference 20 clearly states that Plasmodium strains were contaminated with LDV (see paragraph "P. berghei K173 and P. Yoelii 17X YM stabilates contain LDV virus,..."; also stated in the title of the paper).

2, 129/Sv strain is seldom used for rodent malaria infection. Authors should explain why 129/Sv, not BALB/c or C57BL6, more frequently used for rodent malaria infection, was used in this study.

129/Sv mice were found as a model of choice to analyse endotoxin shock (ref 9), more than BALB/c or C57BL6. This was added in the MM section.

3, In Figure4, “uninfected” should be used instead of “control” in the graph.

Requested change has been made (new Fig 5)

4, The dose of LPS varies among the experiments (200mg or 400mg). Authors should explain how the dose of LPS was determined in each experiment. Moreover, it is common to describe the dose of LPS as mg/kg.

The dose of LPS (200 - 400 µg) that was non lethal for uninfected 129/Sv mice was determined previously (ref 9). We kept the same protocol (including indication of the total dose administered) in this study for consistency. This has been indicated in section 2.2 of the Results.

5, In this study, administration of 400mg LPS did not induce endotoxin shock in mice. The positive control; administration of enough amount of LPS induces endotoxin shock and death of mice should be included in the data. Based on this, sub-lethal or non-lethal dose of LPS should be determined and shown in this study, which could be examined whether susceptibility to endotoxin shock may be increased or not combined with Py infection.

The lethal dose of endotoxin for 129/Sv mice (1000 µg) was determined in Fig 1 of reference 9.

6, Plasmodium “yoelii”, not “yoeli”.

This has been corrected  through the text

7, The parasitemia data is completely lacked in this study. The parasitemia growth should be demonstrated. The reader could not understand the mice condition day 10 after Py infection. Authors should explain why day 10 after Py infection was chosen as timing of administration of LPS.

Parasitemia and anemia data have been added (new Fig. 1) Day 10 p.i. was choosen as the beginning of the most severe phase of infection.

8, Page3 line76-80 should be placed in Figure 2 legend.

Lines 76-80 were supposed to be part of the Figure legend. Typography has been changed to make it clear.

9, Authors should explain how to generate goat anti-TNF serum. Authors cite ref9, but it is not adequate. If this antiserum was used in previously published study, cite that. If not, demonstrate that this antiserum is confirmed to neutralize TNF bioactivity in vivo or in vitro. Is 5mg of anti-TNF serum enough to neutralize TNF activity in mice received LPS? What is control antibody (HIL9K6)? prepared from not immunized goat? Which line indicates mice received control antibody in Figure3?

Anti-TNF antibody was the same as the one used in ref 9. This has now been added. Control antibody was prepared from not immunized goat (added in MM section). Mice receiving control antibody have been indicated in the figure (new Fig 4).

10, Figure4 shows the expression of sCD14 and LBP was increased day10 after Py infection. Why were the expression of these proteins increased? What kind of cells express these proteins? Did authors examine the expression profile (mRNA, protein level) in cells usually expressing these proteins?

More information on the mechanisms of LPS recognition, including the role and origin of sCD14 and LBP has been added in the introduction (with additional references). The cellular expression profile of the soluble LPS receptors was not analysed in this study. Possible mechanisms of induction have been discussed. For LBP, an acute phase protein, inflammation may be the trigger, like in other types of infections. sCD14 production is enhanced by proinflammatory cytokines (references added).

11, The increased level of sCD14 and LBP may be linked to increase susceptibility to endotoxin shock, but it has not been experimentally proven in this study. Is the increase level of sCD14 and LBP enough to increase reactivity of LPS to TLR4? How about the expression level of TLR4 in Py infected mice?

The effect of plasmodium infection on TLR4 expression was not analysed here, but it was in the litterature. This has been discussed with more details. A causative role of soluble receptors would be difficult to prove in vivo (few papers on this topics, discussed in the Discussion section, have conflicting results). Since these receptors appears to be required for LPS response, their neutralization, or their absence would probably lead to complete inhibition of the response to LPS, with or without concomitant Plasmodium infection. We may however propose that they might be use as prognostic markers of susceptibility, since they are elevated in distinct infection models.

Reviewer 2 Report

Comments for paper review

This report presents a short series of experiments in mice investigating if a non-lethal Plasmodium infection (strain P. yoelii 265 BY, Py) increases susceptibility to endotoxin challenge. Their question is a clinically relevant and potentially large problem in populations affected by malaria, where septic shock due to bacterial infection is signficant. The results indicate a robust effect from the infection, and evidence using neutralization antibodies indicate a role for the central proinflammatory cytokine TNF-alpha in determining the increased susceptibility. Additional results showed Py infection caused modest increases in two LPS serum binding proteins than may also mediate these effects, although this idea was not further tested. The overall conclusions of the study were supported by the results, although it was rather narrowly focussed and provides only limited and incremental confirmation of what other much larger studies have already shown.

The following specific points should be addressed (corrections underlined):

Line 11: The geographical area referred is not clear. Please use more accurate region name(s), eg. sub-Saharan Africa, Asia, etc.

Line 14: yoelii and elsewhere

Line 28: The World Health Organisation.

Line 29: Link between first second points is vague. Suggest break this into two sentences.

Line 49: humans, and investigated

Line 55 and parasite infection results: Please present the infection related data (eg. parasitemia) for all the Py infections to demonstrate they were indeed infected and support this statement (line 55). In addition, it’s unclear at what stage of infection the LPS was given and the sCD14 and PBP measures were taken (after peak parasitemia?), or even if the mice were still infected at these points.

Line 59 and Figure 1: Provide the actual length of time. It appears that all Py + LPS mice were deceased within 1 day.

Line 70: Plasmodium in italics.

Figure 2: Several data points are not visible, presumably because the levels were very small. Consider how to best represent these. eg. use log scale, or state levels on the plot.

Line 88 and Figure 3: Which groups are being compared to generate the p-value? Include this information and P-value results in the figure legend.

Figure 3: which mice received control treatment? (indicate on figure key)

Line 107: saline  = control?

Line 124: was

Line 127: rephrase: ‘was responsible to shock”. Eg. .. was responsible for the endotoxic shock

Author Response

We thank the reviewer for his.her useful comments. Changes and answers are as follows:

Line 11: The geographical area referred is not clear. Please use more accurate region name(s), eg. sub-Saharan Africa, Asia, etc.

Appropriate names have been added

Line 14: yoelii and elsewhere

Name has been corrected

Line 28: The World Health Organisation. 

Spelling has been corrected

Line 29: Link between first second points is vague. Suggest break this into two sentences.

The sentence has been split.

Line 49: humans, and investigated

These have been corrected

Line 55 and parasite infection results: Please present the infection related data (eg. parasitemia) for all the Py infections to demonstrate they were indeed infected and support this statement (line 55). In addition, it’s unclear at what stage of infection the LPS was given and the sCD14 and PBP measures were taken (after peak parasitemia?), or even if the mice were still infected at these points.

Parasitemia and anemia data have been added (new Fig. 1) Day 10 p.i. was choosen as the beginning of the most severe phase of infection.

Line 59 and Figure 1: Provide the actual length of time. It appears that all Py + LPS mice were deceased within 1 day.

Indeed, Py + LPS mice died within 1 day. this has been specified in the text.

Line 70: Plasmodium in italics.

Italics have been added

Figure 2: Several data points are not visible, presumably because the levels were very small. Consider how to best represent these. eg. use log scale, or state levels on the plot.

Individual plots have been added (new Fig. 3)

Line 88 and Figure 3: Which groups are being compared to generate the p-value? Include this information and P-value results in the figure legend.

Information and p value have been added in the Figure legend (new Fig. 4)

Figure 3: which mice received control treatment? (indicate on figure key)

Mice receiving control Ab have been indicated in Figure key (new Fig 4)

Line 107: saline = control?

Yes, saline = uninfected controls. This has been added in the legend

Line 124: was

This has been corrected

Line 127: rephrase: ‘was responsible to shock”. Eg. .. was responsible for the endotoxic shock

Sentence has been rephrased

Round 2

Reviewer 1 Report

The revised manuscript has been significantly improved in many points. However, it still needs to be revised. Authors should carefully assign the numbering to cited papers in reference section. ref1 appears twice in 1st version of manuscript.

1, Which is correct as Py infection dose? 106 iRBCs (p2 line63) or 2×106 iRBCs (p6 line218)?

2, According to the data shown in ref12 (Le-Thi-Phuong et al), 129/sv strain is more resistant (NOT sensitive) to endotoxin shock than BALB/c, DBA/2 strain, because LD50 dose (576 microgram) of 129/sv is higher than that of BALB/C, DBA/2 (132, 192 microgram).

3, Figure legends are written in main text again. p4 line116-120, p4 line130-133 in revised manuscript. 

4, Authors may think malaria infection induces expression of sCD14 or LBP, which enhances the susceptibility to endotoxin shock. In this study, time course data of expression level of sCD14 or LBP during Py infection is missing (only day0-uninfected and day10 after infection). For example, in early stage of infection, or after day20 (Py is eliminated), in which would be nearly normal in the level of sCD14 or LBP, the susceptibility to endotoxin shock does not increase? Without these data, the level of sCD14 or LBP during malaria infection would not be proven to be useful as marker of susceptibility to endotoxin shock.

Author Response

We thank the reviewer for his.her careful reviewing and useful comments.

Authors should carefully assign the numbering to cited papers in reference section. ref1 appears twice in 1st version of manuscript.

Reference number have been checked. Ref. 8 was duplicated (with previous 32 which was corrected)

1, Which is correct as Py infection dose? 106 iRBCs (p2 line63) or 2×106 iRBCs (p6 line218)?

The dose was usually 2x106 iRBCs. However, in some of the early experiments such as the one shown in Fig. 1 the dose was 1x106 iRBCs. This has been added in the Material and Method section.

2, According to the data shown in ref12 (Le-Thi-Phuong et al), 129/sv strain is more resistant (NOT sensitive) to endotoxin shock than BALB/c, DBA/2 strain, because LD50 dose (576 microgram) of 129/sv is higher than that of BALB/C, DBA/2 (132, 192 microgram).

Uninfected 129/Sv mice were indeed more resistent to endotoxin shock. However, LDV-infected 129/Sv mice were more sensitive (LD50 dose 11 microgram versus 18 and 38), and therefore, were much more sensitive to the exacerbation of sensitivity by infection (52x for 129/Sv versus 7x for BALB/c and 5x for DBA/2). Our wording was not precise enpough and has been changed (section 4.1. Mice).

3, Figure legends are written in main text again. p4 line116-120, p4 line130-133 in revised manuscript. 

We apologize for these formatting errors that have been corrected

4, Authors may think malaria infection induces expression of sCD14 or LBP, which enhances the susceptibility to endotoxin shock. In this study, time course data of expression level of sCD14 or LBP during Py infection is missing (only day0-uninfected and day10 after infection). For example, in early stage of infection, or after day20 (Py is eliminated), in which would be nearly normal in the level of sCD14 or LBP, the susceptibility to endotoxin shock does not increase? Without these data, the level of sCD14 or LBP during malaria infection would not be proven to be useful as marker of susceptibility to endotoxin shock.

We agree with this comment. The analysis should be further developped in animals, and mostly in patients to clearly show a definitive correlation between the levels of LPS soluble receptors and susceptibility to shock. This hypothesis is however also supported by different published reports of enhanced soluble receptor levels in infection, both in animals and in humans, that lead to enhanced susceptibility to shock. Unfortunately, due to my retirement and the closure of my group, I have no more access to animal facility and no more ethical approval for additional experiments. Therefore, we cannot perform these experiments. This limitation of our study and the need for additional analysis have been more extensively discussed.

Reviewer 2 Report

The authors have satifactorially addressed all the concerns noted in the original review, and this revised version is acceptable for publication.

Author Response

We thank the reviewer for his.her useful help in th revision of this manuscript.